# Comparison of Two Strategies for Hypercholesterolemia Detection through Point-of-Care Testing

**DOI:** 10.3390/diagnostics14020143

**Published:** 2024-01-08

**Authors:** Héctor Eliud Arriaga-Cázares, David Vega-Morales, Carlos Alberto Moreno-Treviño, Juana Lorena Juarez-Juarez, Carlos Azael Pérez-Arizmendi, Alexandro J. Martagón-Rosado

**Affiliations:** 1Instituto Mexicano del Seguro Social, Hospital de Traumatología y Ortopedia N.º 21, Monterrey 64000, Mexico; 2Escuela de Medicina, Instituto Tecnológico y de Estudios Superiores de Monterrey, Monterrey 64849, Mexico; 3Instituto Mexicano del Seguro Social, Hospital General de Zona 17, Monterrey 64420, Mexico; drdavidvega@yahoo.com.mx; 4Instituto Mexicano del Seguro Social, Unidad de Medicina Familiar 26, Monterrey 64117, Mexico; dr.carlosmoreno@gmail.com (C.A.M.-T.); dr.azprz@gmail.com (C.A.P.-A.); 5Institute for Obesity Research, Instituto Tecnologico y de Estudios Superiores de Monterrey, Monterrey 64849, Mexico; 6Unidad de Investigación de Enfermedades Metabólicas, Instituto Nacional de Ciencias Médicas y Nutrición Salvador Zubirán, Ciudad de México 14080, Mexico

**Keywords:** child, mass screening, hypercholesterolemia, dyslipidemia

## Abstract

Background: Childhood dyslipidemia is a common condition that can lead to atherosclerotic cardiovascular disease in adulthood. It is usually multifactorial. Screening for cholesterol disorders in children varies based on risk factors, with some guidelines recommending cascade screening for children with a clear family history of familial hypercholesterolemia, targeted screening for those with specific risk factors, and universal screening. Point-of-care testing (POCT) cholesterol tests offer potential advantages, including ease of use, portability, increased patient access, low cost, fewer medical or laboratory visits, and instant results. This study aimed to evaluate the effect of POCT cholesterol screening on the diagnosis of hypercholesterolemia in children in a family practice setting. Methods: We used a POCT cholesterol analyzer to perform two different (universal and targeted) screening approaches for dyslipidemia in children. We used the NCEP guidelines for the classification of the results. Results: We screened 183 children, 105 in the universal screening group and 78 in the targeted screening group. Eight patients in the targeted screening group had elevated cholesterol levels (*p* = 0.02). Conclusions: All participants received instant feedback and recommendations. Using a targeted screening approach, POCT could be a practical and effective tool for identifying at-risk children with hypercholesterolemia.

## 1. Introduction

Dyslipidemia significantly contributes to the development of atherosclerosis-related coronary artery disease and peripheral vascular disease. These conditions can lead to severe health complications and have a significant impact on overall morbidity. Several interventions have been developed to help manage dyslipidemia and reduce the risk of these diseases [1]. In Mexico, cardiovascular disease is the leading cause of death, accounting for around 20% of all deaths. Among these, 68.5% are due to ischemic heart disease. This alarming statistic highlights the significant impact of heart-related conditions on the population’s health [2].

Lipoproteins are traditionally classified based on size and density as chylomicrons, chylomicron remnants, and very low-density lipoprotein (VLDL) as large and light molecules. In contrast, low-density lipoprotein (LDL) and high-density lipoprotein (HDL) are sequentially smaller and heavier. In humans, LDL particles play a crucial role in transporting cholesterol to peripheral tissues by binding to the LDL receptor, influencing plasma LDL concentrations [3]. Epidemiological evidence consistently links higher concentrations of LDL cholesterol to an increased risk of myocardial infarction and cardiovascular disease. Consequently, reducing LDL cholesterol levels has become a pivotal strategy for preventing cardiovascular diseases [4]. Total and LDL cholesterol concentrations in children vary with age: low at birth, increasing up to 2 years, peaking before puberty, decreasing during adolescence, and rising again during late adolescence and young adulthood. Generally, concentrations are higher in girls and peak approximately one year earlier than in boys [5]. Childhood dyslipidemia is typically multifactorial, except for lipid disorders of genetic origin, such as familial hypercholesterolemia (FH). Childhood hypercholesterolemia is defined as a total cholesterol level above 200 mg/dL or an LDL-C level exceeding 130 mg/dL [6].

Lipid screening in childhood is crucial, aiming to detect dyslipidemia for prompt management through lifestyle modifications and medical treatment. Early intervention significantly reduces the risk of atherosclerotic cardiovascular disease events in adulthood. Therefore, prioritizing childhood lipid screening is essential for preventing future cardiovascular risks and promoting long-term health [7]. While the USPSTF has found inadequate evidence regarding the diagnostic yield distinction between universal and selective screening for familial hypercholesterolemia or multifactorial dyslipidemia, some European countries report positive experiences with cascade screening. However, not all countries have the infrastructure to implement this method [8]. Despite the importance of childhood lipid screening, screening rates in the US persistently remain low, with only 6.6% of children screened from 2009 to 2013 [9].

There are several guidelines for screening cholesterol disorders in children, ranging from avoiding routine screening for those without risk factors and offering targeted screening for those with risk factors, to performing universal screening between the ages of 9 and 11, and repeat screening between 17 and 21 years [10]. According to recent research, a significant number of children with dyslipidemia remain undiagnosed due to screening methods that primarily target overweight children. In fact, nearly half of the children with dyslipidemia who have a normal weight are not identified through current screening practices. Research suggests that screening for lipid disorders in overweight children often misses nearly 50% of children with dyslipidemia who are of normal weight [11]. When screening for dyslipidemia in children, relying solely on risk factor identification may not be sufficient. In fact, studies have shown that this method can miss up to 30–60% of cases. Therefore, it is important to consider other screening methods to accurately identify children with this condition [12]. Regarding screening practices, pediatricians commonly offer universal screening, whereas family medicine physicians tend to prescribe selective screening. Despite the specialty, physicians generally consider it reasonable to screen for dyslipidemia in infancy [13]. Universal lipid screening (ULS) is frequently used to identify children with genetic dyslipidemia, such as familial hypercholesterolemia (FH). However, it can also detect dyslipidemia, elevated triglycerides, and low HDL-C levels resulting from lifestyle factors and obesity [14]. The implementation of ULS among 9–11-year-old children can identify a significant number of children with dyslipidemia, particularly those who might not have been identified through targeted screening [15]. Cascade screening involves identifying a patient with FH and performing active cholesterol testing, genetic testing, or both on all their potentially affected relatives. It is the most widely used method for diagnosing FH, but ineffective. Many affected individuals remain undetected in the community when cascade screening is used as a primary screening method [16]. Targeted screening is provided to individuals at above-average risk who appear healthy, asymptomatic, or unaware of the condition being screened for. This methodical approach searches for individuals with familial hypercholesterolemia among groups of patients with an early development of atherosclerotic vascular lesions [17].

One of the main issues with the screening of hypercholesterolemia in children is the diagnosis of patients with familial hypercholesterolemia. Familial hypercholesterolemia is a condition that is often underdiagnosed and undertreated globally due to a lack of awareness, both among the public and healthcare professionals. The prevalence of FH in the general population is unknown in 90% of countries in the world [18]. The Netherlands and Norway both have effective national screening programs for familial hypercholesterolemia (FH). As of 2020, it is estimated that Norway has identified 51.33% of its FH population, while the Netherlands identified 41.49% of its FH population as of 2014. In comparison, England identified only 3.98% of its population as having FH from 2003 to 2018 [19]. It is estimated that less than 1% of individuals with FH in the United States have been identified [20]. Data are scarce in Latin America regarding models of care, screening strategies, cost, treatment effectiveness, morbidity, and mortality for hypercholesterolemia. This information is crucial for the development of tools for early diagnosis and treatment, raising awareness of the disease among carriers, family members, and health authorities, and promoting genetic and clinical research specific to Latin American populations, with the implementation of a defined methodology for cascade screening to identify affected individuals [21]. To prevent cardiovascular disease, adopting a preventive medicine approach is crucial, where early detection through global screening and low-cost therapeutic intervention is implemented in primary care [22].

Cholesterol point-of-care testing (POCT) offers several potential advantages as a screening strategy, including ease of use, portability, increased patient access, low cost, fewer doctor or laboratory visits, and instant results [23]. POCT refers to biological measurements determined outside the laboratory, near the patient’s location, and carried out by personnel not directly involved in patient care [24]. The POCT process includes three stages: preanalytical, analytical, and postanalytical. During the preanalytical phase, sample preparation takes place. The analytical phase is the stage at which the test sequence is performed using standard test strips that consist of a porous matrix with embedded dried sections supported by an element. Changes occur when the bar layer penetrates and soaks in [25]. The postanalytical phase begins after the test is completed and the result is obtained, during which the test results are communicated to the patient or health team, allowing for appropriate actions and interventions in case of an abnormal result [26]. A systematic review revealed that the screening and management of familial hypercholesterolemia in pediatric populations was cost-effective, regardless of the age of the children. However, cost-effectiveness varied by case identification method: targeted screening was generally less costly overall, but less effective than population-wide screening, although both methods were generally considered cost-effective [27]. While POCT is as effective as laboratory-based testing for several analytes with high consumer satisfaction, its long-term cost-effectiveness remains a topic of debate. However, conflicting findings regarding the overall cost-effectiveness of POCT have also been reported, highlighting the need for further investigation and analysis. [28].

This study aimed to evaluate the impact of POCT cholesterol screening on the diagnosis of hypercholesterolemia in children within a family practice setting.

## 2. Materials and Methods

This study aimed to evaluate the effectiveness of two screening strategies for detecting hypercholesterolemia using POCT. The study was conducted at the Unidad de Medicina Familiar 26, a family practice clinic at the Instituto Mexicano del Seguro Social in Monterrey, Mexico, from September 2021 to June 2022. Screening strategies included targeted and universal screening.

### 2.1. Study Participants

We aimed to pinpoint the offspring of parents who may have familial hypercholesterolemia in the family medicine outpatient clinic and via electronic records, using the criteria outlined in the questionnaire as our guide [29]. Once these potential participants were identified, we intentionally checked whether they had pediatric-aged children. Subsequently, we asked if they had children and requested to undergo a POCT cholesterol test. The criteria for this supplementary search are based on the questions outlined in Table 1.

For the universal screening strategy, children attending the family practice clinic on designated study days were approached by a fellow researcher in the waiting room and invited to participate in the study. During the selection process, parents were queried about any existing comorbidities, and individuals at risk for dyslipidemia were excluded. Patients who met the inclusion criteria and provided informed consent underwent cholesterol measurement in the waiting room.

Demographic characteristics (age, sex, and parent-reported health history), clinical data (month and year of visit, body mass index (BMI), BMI percentile, and family history of CVD, questionnaire), and lipid screening were obtained in the clinic. Weight status was determined based on the Centers for Disease Control and Prevention guidelines, with BMI < 85th percentile characterized as being normal, BMI in the 85th to 94th percentile as overweight, and BMI ≥ 95th percentile as obese. The results of the screening were categorized into three groups based on their cholesterol screening levels according to the National Cholesterol Education Program (NCEP) guidelines [30] for children and adolescents. Specifically, we classified cholesterol levels as follows: acceptable < 170 mg/dL, borderline if it was between 170 and 199 mg/dL, and elevated if it was higher than 200 mg/dL. The decision to focus solely on cholesterol measurement was in accordance with the recommendations of the NCEP. This approach aligns with the ‘Make Early Diagnosis to Prevent Early Deaths’ criteria for diagnosis of familial hypercholesterolemia [31] using age, family history, and total cholesterol (Table 2). If total cholesterol levels indicated a potential genetic disorder, a complete lipid panel would be conducted. We excluded patients with any disease that can cause secondary dyslipidemia.

### 2.2. Measurement of Cholesterol

Total cholesterol was measured with capillary blood samples collected through transcutaneous puncture on the medial side of the tip of the index finger using a disposable hypodermic lancet. Before puncture, 70% alcohol was applied to skin to promote antisepsis. The first drop of blood was discarded, and the following were used in random order for the analyses. Next, the blood sample was applied to each device’s test strip within 10 s. The measurement of cholesterol was performed with the Accutrend Plus system (Roche Diagnostics, Mannheim, Germany). The monitor is highly concordant with the laboratory results (Lin’s coefficient = 0.944) [32], and the total cholesterol results were delivered in three minutes. All blood samples were obtained between 0700 and 1100 h in a fasting state. Patients with diabetes mellitus, kidney disease, or a previous diagnosis of dyslipidemia were excluded from the study.

### 2.3. Statistical Analyses

Parametric variables were presented as means and standard deviations, and nonparametric values were presented as medians and interquartile ranges. Continuous variables that satisfied normality were compared using the *t*-test. The chi-square test was used to analyze categorical demographic data. Differences were considered significant when *p* < 0.05. Data analysis was performed using SPSS version 23.

### 2.4. Ethical Considerations

The study was conducted after obtaining an ethics approval from the 1903 Local Research and Ethics Committee of the Instituto Mexicano del Seguro Social. The trial followed recommendations for Good Clinical Practice, the Declaration of Helsinki, and local laws. All patients (or their parents or legal guardians) provided written informed consent for participation.

## 3. Results

A total of 183 children were screened between September 2021 and June 2022. Lipid screening was performed in 105 children in the ULS group. The remaining 78 patients were included in the targeted screening group (Table 3).

The proportion of the males was 50.5% in the ULS group and 42% in the targeted screening group, respectively (*p* = 0.27). The mean age in the ULS group was 8.4 ± 3.8 years and 7.6 ± 4.3 years in the targeted screening group (*p* = 0.19)

Out of the targeted screening group and the ULS group, 59% and 61% of the subjects, respectively, had a normal BMI. The number of children with obesity in the targeted screening group was 17 (22%), while the ULS group had 19 (18%). The difference between the two groups was not statistically significant (*p* = 0.28).

In the ULS group, 93 (85%) patients had a family history of disease. The most common antecedent was diabetes mellitus in four parents (5%). As for the targeted screening group, the family history was as follows: 42 (53%) had a father’s disease, 32 (42%) had a maternal disease, while in the remaining 4 (5%), both parents were affected.

In the ULS group, 100 out of 105 participants (95%) had normal cholesterol levels while 5 (5%) had borderline levels. None of the subjects had cholesterol levels higher than 200 mg/dL, and the maximum cholesterol level in this group was 190 mg/dL. On the other hand, in the targeted screening group, 69 out of 78 patients (88%) had normal cholesterol levels, 1 (1%) had borderline levels, and 8 (10%) had levels higher than 200 mg/dL (*p* = 0.02). The highest cholesterol level observed in this group was 233 mg/dL. In accordance with the MEDPED guidelines, we conducted a total lipid assessment for this patient. The findings revealed a total cholesterol level of 165 mg/dL and an LDL-cholesterol level of 120.5 mg/dL. We recommended a nutritional evaluation and a follow-up lipid panel in six months.

## 4. Discussion

The conducted study has revealed the significant potential of POCT cholesterol testing in detecting susceptible young individuals. Children with a family history of lipid disorders and coronary artery disease are more vulnerable to developing such conditions. Therefore, it is crucial to consider the patient’s family history when performing selective screening, as it helps identify those who are at a higher risk for developing lipid disorders and eventually coronary artery disease.

It was found in our study that hypercholesterolemia was prevalent in 4% of the children, which is close to the 5.2–6.6% prevalence reported in the National Health and Nutrition Examination Survey (NHANES) conducted in the United States [33]. As a result, we have found that our screening results are generally consistent with those of other screening programs, but with the added advantage of immediate feedback for the patient. By providing efficient screening and referral for dyslipidemia, POCT may have the potential to improve patient outcomes by providing an effective means of screening and referral. It is critical to recognize, however, that further studies are required in order to establish best practices and to develop successful reimbursement models in the future [34].

After carrying out our study, we discovered that the group of patients who underwent targeted screening had a significantly higher number of cases with elevated cholesterol levels in comparison to the group that underwent universal screening. As a result of the targeted screening strategy, 10% of the patients had elevated cholesterol levels. This finding is consistent with figures reported in the ‘Screening for Lipid Disorders in Children and Adolescents’ study [35], where the prevalence of elevated total cholesterol levels (≥200 mg/dL) ranged from 7.1% to 9.4%. Based on the findings of this research, it can be inferred that the selected strategy can achieve the desired outcomes and can be considered a viable option for future research studies. The extent to which this occurs will largely depend on the cultural and geographical context of the population in question.

Childhood might be the optimal period for cholesterol screening, as screening in early childhood by family practice physicians can successfully detect dyslipidemia, including its primary forms. The mean age of the individuals who underwent screening was eight years, slightly below the recommended age of nine as suggested by the NCEP guidelines [36]. However, there are alternative approaches that advocate commencing screening at a younger age [37,38]. We believe that in our population, this younger age group often seeks medical attention for routine childhood ailments or vaccinations, making it more practical to conduct the screening process during such visits.

Childhood obesity is becoming an epidemic, and it is a critical issue affecting children’s health and increasing the risk for cardiovascular outcomes. The BMI of our study participants aligned with that of the general population, as reported in the recent ENSANUT report in Mexico [39]. Obese patients had higher cholesterol levels, which is consistent with previous research [40]. It is important to note that while obesity increases the risk of dyslipidemia, some normal-weight children may also have lipid abnormalities that can be identified through targeted screening [41].

By incorporating the questionnaire as a tool for the target screening group, a family history of parental hypercholesterolemia emerged as a more reliable predictor of disease. This translates into a more effective screening strategy, as reported by Robledo [42]. Targeted screening in specific population or geographic regions can help identify clusters of individuals with familial hypercholesterolemia, which may benefit those in areas with a high prevalence of the disease. These findings have the potential to enhance screening practices compared to standard approaches [43]. In a study aimed to determine the costs and benefits of different screening strategies, a cost-effectiveness analysis of different screening strategies for FH was conducted in a simulated population in England and Wales aged 16–54 years. [44]. The strategies considered were universal screening, opportunistic screening in primary care, screening of patients admitted to the hospital with premature myocardial infarction, and tracing family members of affected patients. The results showed that the tracing of family members was the most cost-effective strategy and that this strategy required screening 2.6 individuals to identify 1 case. Universal population screening was the least cost-effective strategy, and screening 1365 individuals was required to detect 1 case. Screening younger people and women was more cost-effective for each strategy. On the other hand, a study aimed at assessing the sensitivity and specificity of family history in identifying children with severe or genetic hyperlipidemia found that using family history to selectively determine the need for cholesterol screening would have missed many children with moderate hyperlipidemia and failed to detect a substantial number of children who likely had familial hypercholesterolemia and required pharmacological treatment. This study recommends that universal cholesterol screening in the pediatric population will allow the early diagnosis and appropriate treatment of children with significant dyslipidemia secondary to genetic and/or adverse lifestyle factors, hopefully preventing arterial disease [45]. The *European Journal of Preventive Cardiology* stated that screening for familial hypercholesterolemia in children should be country-specific, utilizing all existing screening strategies, including opportunistic screening in the setting of a positive family history [46]. The process of cascade screening for dyslipidemia involves testing the cholesterol levels of close relatives of individuals who meet either genetic or phenotypic criteria for a diagnosis of familial hypercholesterolemia. Approximately half of first-degree relatives of affected subjects with the disease will also have it [47]. In Spain, cascade screening has led to the earlier identification of FH and improved survival rates [48]. By initiating pharmacologic treatment, life expectancy has been extended, resulting in a cost-saving advantage for the national screening program. However, laboratories in family care clinics are capable of screening for individuals with potential lipid disorders in routine tests and identifying those at risk, focusing on finding those with an LDL-cholesterol cut-off point of 250 mg/dL, regardless of age [49]. Previous research suggests that this approach can be effective in identifying adult patients with lipid disorders, finding a prevalence of familial hypercholesterolemia like that reported in the general population [50].

Although the traditional belief is that central laboratory testing is less expensive than POCT, the involvement of laboratory staff may render it more costly and infeasible in low-income settings [51]. In the context of developing countries, it may be worthwhile to consider investing in POCT as the start-up costs for infrastructure are relatively lower, making it a more cost-effective option in primary care settings [52]. In Australia, the costs associated with POCT for lipids were found to be higher for the healthcare sector, but this difference was not statistically significant. Despite this, POCT resulted in significant cost savings for patients and their families by achieving reductions in total cholesterol after the introduction of POCT. [53]. The most convincing argument for POCT is in regions where the clinical laboratory is far away, such as remote rural areas. In some countries, the distance can be great. With POCT available, decisions can be made immediately, eliminating the need to send samples to the clinical laboratory and reducing the risk of preanalytical errors [14]. Additionally, POCT allows for efficient patient triage for transfer to major medical centers. In cases where monitoring treatment is necessary, the capability to remotely measure values would offer an advantageous opportunity [25]. POCT can expedite clinical diagnosis and enhance patient-centered outcomes in resource-limited settings where laboratory facilities and trained personnel are scarce. As POCT are increasingly being developed for use in low- and middle-income countries, the approach to evaluating their effectiveness will differ from that of this approach in developed countries. Thus, it is crucial to rigorously assess POC tests for the patient-centered outcomes they are intended to address and in the context for which they were designed [54].

The efficacy of screening asymptomatic children for dyslipidemia and managing it early through lifestyle changes or medication in delaying or reducing the incidence of myocardial infarction or stroke in adulthood is still uncertain [55]. Additionally, there is a need for improvements in several areas, including increasing awareness and advocacy, establishing specialized centers for the diagnosis and management of familial hypercholesterolemia, developing family-based care plans, and implementing country-specific advocacy organizations to increase FH awareness and screening. Furthermore, improvements in screening, testing, diagnosis, treatment, and registries are necessary [56].

In this context, POCT is crucial for swift identification, risk mitigation, and the prevention of potential medical concerns. POCT stands out for its ability to deliver rapid and precise results, empowering healthcare professionals to take immediate actions, such as prescribing medications or recommending lifestyle changes. This enables the prevention or management of lipid disorders and coronary artery disease. The overarching goal is to swiftly address health concerns through the quick and accurate information provided by POCT, thereby enhancing overall patient care.

### Study Limitations

This study was conducted at a single family-practice clinic in Monterrey, Mexico, which limits the generalizability of the findings to broader populations and different healthcare settings. Therefore, the results may not be universally applicable, especially considering cultural and regional variations in health practices.

To conduct the universal screening strategy, the researchers approached children attending the family practice clinic on designated days, which could introduce a selection bias, as those attending on specific days might differ from the broader population. Cholesterol measurements were obtained through POCT, which is efficient but may have limitations compared to more comprehensive laboratory-based assessments, potentially impacting the precision of cholesterol measurements.

The efficacy of the questionnaire in identifying familial hypercholesterolemia cases depends on the accuracy of responses and the reliability of family history information. However, these factors may be subject to recall bias, which could affect the accuracy of the results. Interpreting serum lipid profiling findings and developing clinical practice and policy recommendations requires the careful consideration of several limitations, including family history, variations, and inaccuracies.

A significant weakness in the current research is the absence of genetic analysis. Genetic variables are known to play a pivotal role in the emergence of dyslipidemia and hypercholesterolemia, making it critical to understand the inherited risks faced by children in communities with a history of dyslipidemia. A more profound comprehension of these inherited variables may be attained by exploring biological variables, including genetic testing, in subsequent studies aimed at comprehensively investigating the genetic causes of hypercholesterolemia in children. Advancing our knowledge in these areas is fundamental to enhancing our understanding of childhood dyslipidemia and developing effective screening strategies.

## 5. Conclusions

The results of our study showed that the targeted questionnaire screening method was significantly more effective in identifying children with hypercholesterolemia than the universal approach. This was particularly true in settings with limited resources, where universal screening may not be feasible due to cost constraints. We hope that our findings will encourage further research in this area and lead to the development of more effective screening methods for children at risk of hypercholesterolemia.

## Figures and Tables

**Table 1 diagnostics-14-00143-t001:** Questionnaire for targeted screening group.

1.-Do the biological mother or father have elevated blood lipid levels, or does anyone take cholesterol-lowering medications (statins)?
2.-Do the biological mother or father have fatty lumps on the skin (xanthomas), particularly in the Achilles tendon/hand/knee or eye areas (xanthelasma)?
3.-Has the biological mother or father experienced a heart attack or stroke before the age of 55?

**Table 2 diagnostics-14-00143-t002:** The US (MEDPED) diagnostic criteria for familial hypercholesterolemia (FH).

Age (Years)	First Degree Relative with FH	Second Degree Relative with FH	Third Degree Relative with FH	General Population
<20	220	230	240	270
20–29	240	250	260	290
30–39	270	280	290	340
≥40	290	300	310	360

FH is diagnosed if total cholesterol exceeds these cut-points.

**Table 3 diagnostics-14-00143-t003:** Population characteristics and results of 183 children screened for hypercholesterolemia.

		Universal Screening (*N* = 105)	Targeted Screening (*N* = 78)	*p*-Value
Age, years ± mean (SD)		8.4 ± 3.8	7.6 ± 4.3	0.19
Sex—*N* (%)				0.27
	Male Female	53 (50.5) 52 (49.5)	33 (42) 45 (58)	
BMI ^α^ classification—N(%)				0.28
	Normal	64 (61)	46 (59)	
	Overweight	22 (21)	15 (19)	
	Obesity	19 (18)	17 (22)	
Cholesterol levels—N(%)				0.02
	Normal	100 (95)	69 (89)	
	Borderline	5 (5)	1 (1)	
	Elevated	0 (0)	8 (10) ^β^	

^α^ BMI = Body mass index; ^β^ indicates a statistically significant difference between the groups.

## Data Availability

Dataset available on request from the authors.

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
