# Peer review of "Comparison of Two Strategies for Hypercholesterolemia Detection through Point-of-Care Testing"

_diagnostics, 2024, doi:10.3390/diagnostics14020143_

Round 1

Reviewer 1 Report

Comments and Suggestions for Authors

Early detection and treatment of the hypercholesterolemia in childhood and adolescence are critical for increasing life expectancy. The authors proposed an interesting approach to the diagnosis of hypercholesterolemia in childhood using Point of care testing (POCT) cholesterol tests.

Major Comments:
1. The study design is somewhat questionable. In the articles cited by the authors [23,24], screening was carried out not only for total cholesterol, but also for low-density lipoprotein cholesterol and non-HDL cholesterol. Will measuring total cholesterol be sufficient to detect hypercholesterolemia?
It would be advisable to conduct additional studies of low-density lipoprotein cholesterol and non-HDL cholesterol in the universal screening and targeted screening groups and compare the likelihood of detecting hypercholesterolemia when POCT alone is performed. This may exclude underdiagnosis of hypercholesterolemia.

2. The Discussion section should include the limitations of the study.

Minor Comments:

Have you checked triglyceride levels in children? Usually the Accutrend Plus system allows you to do this.

Author Response

Dear Reviewer

Please see the attachment for the reply to your comments. 

Kind regards. 

Reviewer 2 Report

Comments and Suggestions for Authors

The authors in this study attempt to investigate the applicability of a point-of-care test (POCT) for the diagnosis of hypercholesterolemia in children. The Accutrend Plus system was used for this purpose where they tested 183 children – 105 of them were assigned to the universal lipid screening (ULS) group and 78 were assigned to the targeted screening group.

The current experimental design lacks clarity, and the conclusions drawn are not sufficiently substantiated by the conducted study. To address these issues, significant additional work or a complete restructuring of the paper is necessary. Detailed suggestions and specific areas requiring attention are outlined in the comments below:

Major concerns:

The current experimental design of the study presents ambiguity regarding its primary objective. It is not evident whether the study aims to compare the ULS method with the targeted screening approach, or if it intends to assess the effectiveness of the POCT for hypercholesterolemia.

 If the goal is to compare ULS and targeted screening, the study should use the same patient group for both screening methods, tracking, and comparing outcomes to evaluate the effectiveness of each approach in diagnosing hypercholesterolemia. This would involve a side-by-side comparison of how well each method identifies patients with the condition and can clearly quantify the impact a screening questionnaire can have on identifying at-risk patients.

 On the other hand, if the objective is to evaluate the efficacy of the POCT, the study should involve testing the same patients using both the POCT and a conventional laboratory-based assay. This comparison would focus on aspects such as diagnostic accuracy, ease of use, time to deliver results, and cost-effectiveness of the two different testing methods.

Currently, the study has screened two distinct patient populations using different approaches (ULS and targeted screening) solely with the Accutrend Plus POCT platform. With this setup, the study falls short in providing a meaningful comparison between the two screening methods or a comprehensive evaluation of the POCT’s efficacy. To draw valid conclusions, a redesign of the study is required to align with one of the two outlined objectives and ensure an effective and accurate comparison.

Minor concerns:

1.     The questionnaire detailed in Section 2.1 would be better presented as a separate exhibit, specifically in a table format, rather than being included within the main body of the text.

2.     Spelling error in Section 2.1, second line. It should be “POCT” and not “PCOT”

Author Response

(The authors gave the same response as above.)

Round 2

Reviewer 1 Report

Comments and Suggestions for Authors

The authors have provided sufficient background, methodology, results and discussion. The paper was also supplemented with relevant citations.

 Minor corrections:

Page 5. Line 206: “We excluded patients with any disease that may cause dyslipidemia.”

Do you mean the secondary dyslipidemia?

Author Response

Dear reviewer.

Please see the attachment with the point-by-point response to the your comments. 

Thank you for your time

Reviewer 2 Report

Comments and Suggestions for Authors

The authors have effectively enhanced the methods and discussion sections of their paper, providing a clear description and differentiation of various hypercholesterolemia screening methods. However, the misalignment between the study's experimental design and its stated conclusions has still not been resolved, indicating a need for more precise focus. Despite the authors’ assertion in their response letter that both screening approaches targeted the same population (i.e., children), the specific patient groups compared in each approach were distinct. To conduct a comprehensive comparison of the efficacy and cost-effectiveness of the targeted screening approach, utilizing the same group of patients for both screening methods would have been preferable. Such a design would have definitively established the effectiveness of the questionnaire in identifying at-risk patients more precisely.

As the paper currently stands, the application of different screening approaches to separate groups of children impedes a direct comparison of the methods’ outcomes. A more viable approach might involve a detailed exploration of just the targeted screening method. By focusing on this approach, the paper could elucidate and quantify the potential benefits in time and cost savings achieved by screening fewer patients—specifically, those identified as at-risk. This concentrated examination could offer more explicit insights into the advantages of targeted screening for the detection of hypercholesterolemia in pediatric patients. In its present form, the study falls short of providing a thorough comparative analysis between the two screening methods.

Author Response

(The authors gave the same response as above.)

Round 3

Reviewer 2 Report

Comments and Suggestions for Authors

Authors have addressed the concerns raised by including further clarification in the conclusion and introduction sections.